# Skip Connections Eliminate Singularities

**A. Emin Orhan**
aeminorhan@gmail.com

**Xaq Pitkow**
xaq@rice.edu

Baylor College of Medicine & Rice University

## Abstract

Skip connections made the training of very deep networks possible and have become an indispensable component in a variety of neural architectures. A completely satisfactory explanation for their success remains elusive. Here, we present a novel explanation for the benefits of skip connections in training very deep networks. The difficulty of training deep networks is partly due to the singularities caused by the non-identifiability of the model. Several such singularities have been identified in previous works: (i) overlap singularities caused by the permutation symmetry of nodes in a given layer, (ii) elimination singularities corresponding to the elimination, i.e. consistent deactivation, of nodes, (iii) singularities generated by the linear dependence of the nodes. These singularities cause degenerate manifolds in the loss landscape that slow down learning. We argue that skip connections eliminate these singularities by breaking the permutation symmetry of nodes, by reducing the possibility of node elimination and by making the nodes less linearly dependent. Moreover, for typical initializations, skip connections move the network away from the "ghosts" of these singularities and sculpt the landscape around them to alleviate the learning slow-down. These hypotheses are supported by evidence from simplified models, as well as from experiments with deep networks trained on real-world datasets.

## 1 Introduction

Skip connections are extra connections between nodes in different layers of a neural network that skip one or more layers of nonlinear processing. The introduction of skip (or residual) connections has substantially improved the training of very deep neural networks (He et al., 2015; 2016; Huang et al., 2016; Srivastava et al., 2015). Despite informal intuitions put forward to motivate skip connections, a clear understanding of how these connections improve training has been lacking. Such understanding is invaluable both in its own right and for the possibilities it might offer for further improvements in training very deep neural networks. In this paper, we attempt to shed light on this question. We argue that skip connections improve the training of deep networks partly by eliminating the singularities inherent in the loss landscapes of deep networks. These singularities are caused by the non-identifiability of subsets of parameters when nodes in the network either get eliminated (*elimination singularities*), collapse into each other (*overlap singularities*) (Wei et al., 2008), or become linearly dependent (*linear dependence singularities*). Saad & Solla (1995); Amari et al. (2006); Wei et al. (2008) identified the elimination and overlap singularities and showed that they significantly slow down learning in shallow networks; Saxe et al. (2013) showed that linear dependence between nodes arises generically in randomly initialized deep linear networks and becomes more severe with depth. We show that skip connections eliminate these singularities and provide evidence suggesting that they improve training partly by ameliorating the learning slow-down caused by the singularities.

## 2 Results

### 2.1 Singularities in fully-connected layers and how skip connections break them

In this work, we focus on three types of singularity that arise in fully-connected layers: elimination and overlap singularities (Amari et al., 2006; Wei et al., 2008), and linear dependence singularities

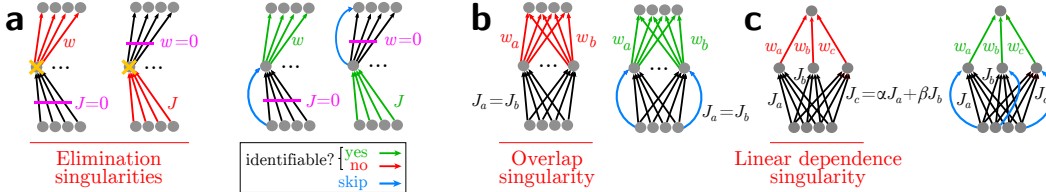

Figure 1: Singularities in a fully connected layer and how skip connections break them. **(a)** In elimination singularities, zero incoming weights, $J = 0$, eliminate units and make outgoing weights, $w$, non-identifiable (red). Skip connections (blue) ensure units are active at least sometimes, so the outgoing weights are identifiable (green). The reverse holds for zero outgoing weights, $w = 0$: skip connections recover identifiability for $J$. **(b)** In overlap singularities, overlapping incoming weights, $J_a = J_b$, make outgoing weights non-identifiable; skip connections again break the degeneracy. **(c)** In linear dependence singularities, a subset of the hidden units become linearly dependent, making their outgoing weights non-identifiable; skip connections break the linear dependence.

(Saxe et al., 2013). The linear dependence singularities can arise exactly only in linear networks, whereas the elimination and overlap singularities can arise in non-linear networks as well. These singularities are all related to the non-identifiability of the model. The Hessian of the loss function becomes singular at these singularities (Supplementary Note 1), hence they are sometimes also called degenerate or higher-order saddles (Anandkumar & Ge, 2016). Elimination singularities arise when a hidden unit is effectively killed, e.g. when its incoming (or outgoing) weights become zero (Figure 1a). This makes the outgoing (or incoming) connections of the unit non-identifiable. Overlap singularities are caused by the permutation symmetry of the hidden units at a given layer and they arise when two units become identical, e.g. when their incoming weights become identical (Figure 1b). In this case, the outgoing connections of the units are no longer identifiable individually (only their sum is identifiable). Linear dependence singularities arise when a subset of the hidden units in a layer become linearly dependent (Figure 1c). Again, the outgoing connections of these units are no longer identifiable individually (only a linear combination of them is identifiable).

How do skip connections eliminate these singularities? Skip connections between adjacent layers break the elimination singularities by ensuring that the units are active at least for some inputs, even when their adjustable incoming or outgoing connections become zero (Figure 1a; right). They eliminate the overlap singularities by breaking the permutation symmetry of the hidden units at a given layer (Figure 1b; right). Thus, even when the adjustable incoming weights of two units become identical, the units do not collapse into each other, since their distinct skip connections still disambiguate them. They also eliminate the linear dependence singularities by adding linearly independent (in fact, orthogonal in most cases) inputs to the units (Figure 1c; right).

## 2.2 WHY ARE SINGULARITIES HARMFUL FOR LEARNING?

The effect of elimination and overlap singularities on gradient-based learning has been analyzed previously for shallow networks (Amari et al., 2006; Wei et al., 2008). Figure 2a shows the simplified two hidden unit model analyzed in Wei et al. (2008) and its reduction to a two-dimensional system in terms of the overlap and elimination variables, $h$ and $z$. Both types of singularity cause degenerate manifolds in the loss landscape, represented by the lines $h = 0$ and $z = \pm1$ in Figure 2b, corresponding to the overlap and elimination singularities respectively. The elimination manifolds divide the overlap manifolds into stable and unstable segments. According to the analysis presented in Wei et al. (2008), these manifolds give rise to two types of plateaus in the learning dynamics: *on-singularity plateaus* which are caused by the random walk behavior of stochastic gradient descent (SGD) along a stable segment of the overlap manifolds (thick segment on the $h = 0$ line in Figure 2b) until it escapes the stable segment, and (more relevant in practical cases) *near-singularity plateaus* which manifest themselves as a general slowing of the dynamics near the overlap manifolds, even when the initial location is not within the basin of attraction of the stable segment.

Although this analysis only holds for two hidden units, for higher dimensional cases, it suggests that overlaps between hidden units significantly slow down learning along the overlap directions. These overlap directions become more numerous as the number of hidden units increases, thus reducing the effective dimensionality of the model. We provide empirical evidence for this claim below.

Figure 2: Why singularities are harmful for gradient-based learning. (**a**) Diagram of the analyzed network and parameter reduction performed in the analysis in Wei et al. (2008). $h = 0$ corresponds to the overlap singularity and $z = \pm 1$ correspond to the elimination singularities. (**b**) The gradient flow field for the two-dimensional reduced system. The gradient norm is indicated by color. The segment marked by the thick solid line is stable in this example; its basin of attraction is shaded in gray. (**c**) Near-singularity plateau. Trajectory of learning dynamics starting from the black dot indicated in **b**. Analysis and plots adapted from Wei et al. (2008). (**d**) Illustration of a linear dependence manifold in a toy model: the new coordinate $m$ represents the distance to a particular linear dependence manifold. (**e**) Gradient flow field for the toy model shown in **d**.

As mentioned earlier, linear dependence singularities arise exactly only in linear networks. However, we expect them to hold approximately, and thus have consequences for learning, in the non-linear case as well. Figure 2d-e shows an example in a toy single-layer nonlinear network: learning along a linear dependence manifold, represented by $m$ here, is much slower than learning along other directions, e.g. the norm of the incoming weight vector $J_c$ in the example shown here. Saxe et al. (2013) demonstrated that this linear dependence problem arises generically, and becomes worse with depth, in randomly initialized deep linear networks. Because learning is significantly slowed down along linear dependence directions compared to other directions, these singularities effectively reduce the dimensionality of the model, similarly to the overlap manifolds.

### 2.3 PLAIN NETWORKS ARE MORE DEGENERATE THAN NETWORKS WITH SKIP CONNECTIONS

To investigate the relationship between degeneracy, training difficulty and skip connections in deep networks, we conducted several experiments with deep fully-connected networks. We compared three different architectures. (i) The *plain* architecture is a fully-connected feedforward network with no skip connections, described by the equation:

$$\mathbf{x}_{l+1} = f(\mathbf{W}_l \mathbf{x}_l + \mathbf{b}_{l+1}) \qquad l = 0, \dots, L-1$$

where $f$ is the ReLU nonlinearity and $\mathbf{x}_0$ denotes the input layer. (ii) The *residual* architecture introduces identity skip connections between adjacent layers (note that we do not allow skip connections from the input layer):

$$\mathbf{x}_1 = f(\mathbf{W}_0 \mathbf{x}_0 + \mathbf{b}_1), \quad \mathbf{x}_{l+1} = f(\mathbf{W}_l \mathbf{x}_l + \mathbf{b}_{l+1}) + \mathbf{x}_l \quad l = 1, \dots, L-1$$

(iii) The *hyper-residual* architecture adds skip connections between each layer and all layers above it:

$$\mathbf{x}_1 = f(\mathbf{W}_0\mathbf{x}_0+\mathbf{b}_1), \ \mathbf{x}_2 = f(\mathbf{W}_1\mathbf{x}_1+\mathbf{b}_2)+\mathbf{x}_1, \ \mathbf{x}_{l+1} = f(\mathbf{W}_l\mathbf{x}_l+\mathbf{b}_{l+1})+\mathbf{x}_l+\frac{1}{l-1}\sum_{k=1}^{l-1}\mathbf{Q}_k\mathbf{x}_k \quad l = 2, \dots, L-1$$

The skip connectivity from the immediately preceding layer is always the identity matrix, whereas the remaining skip connections $\mathbf{Q}_k$ are fixed, but allowed to be different from the identity (see Supplementary Note 2 for further details). This architecture is inspired by the DenseNet architecture (Huang et al., 2016). In both architectures, each layer projects skip connections to layers above it. However, in the DenseNet architecture, the skip connectivity matrices are learned, whereas in the hyper-residual architecture considered here, they are fixed.

In the experiments of this subsection, the networks all had $L = 20$ hidden layers (followed by a softmax layer at the top) and $n = 128$ hidden units (ReLU) in each hidden layer. Hence, the networks had the same total number of parameters. The biases were initialized to 0 and the weights were initialized with the Glorot normal initialization scheme (Glorot & Bengio, 2010). The networks were trained on the CIFAR-100 dataset (with coarse labels) using the Adam optimizer (Kingma &

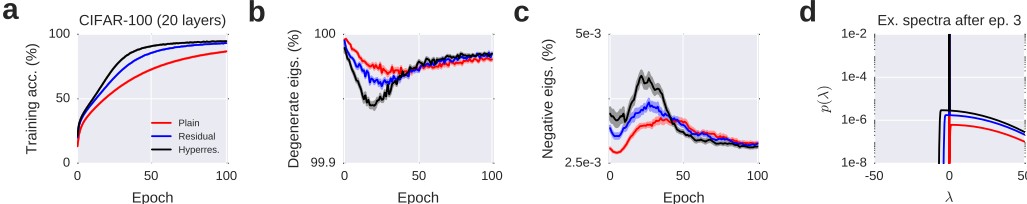

Figure 3: Model degeneracy increases training difficulty. (**a**) Training accuracy of different architectures. (**b**) Estimated fraction of degenerate eigenvalues during training. Error bars are standard errors over 50 independent runs of the simulations. (**c**) Estimated fraction of negative eigenvalues during training. (**d**) Example fitted spectra after 3 training epochs.

Ba, 2014) with learning rate 0.0005 and a batch size of 500. Because we are mainly interested in understanding how singularities, and their removal, change the shape of the loss landscape and consequently affect the optimization difficulty, we primarily monitor the training accuracy rather than test accuracy in the results reported below.

To measure degeneracy, we estimated the eigenvalue density of the Hessian during training for the three different network architectures. The probability of small eigenvalues in the eigenvalue density reflects the dimensionality of the degenerate parameter space. To estimate this eigenvalue density in our $\sim$ 1M-dimensional parameter spaces, we first estimated the first four moments of the spectral density using the method of Skilling (Skilling, 1989) and fit the estimated moments with a flexible mixture density model (see Supplementary Note 3 for details) consisting of a narrow Gaussian component to capture the bulk of the spectral density, and a skew Gaussian density to capture the tails (see Figure 3d for example fits). From the fitted mixture density, we estimated the fraction of degenerate eigenvalues and the fraction of negative eigenvalues during training.

We validated our main results, as well as our mixture model for the spectral density, with smaller networks with $\sim$ 14K parameters where we could calculate all eigenvalues of the Hessian numerically (Supplementary Note 4). For these smaller networks, the mixture model slightly underestimated the fraction of degenerate eigenvalues and overestimated the fraction of negative eigenvalues; however, there was a highly significant linear relationship between the actual and estimated fractions.

Figure 3b shows the evolution of the fraction of degenerate eigenvalues during training. A large value at a particular point during optimization indicates a more degenerate model. By this measure, the hyper-residual architecture is the least degenerate and the plain architecture is the most degenerate. We observe the opposite pattern for the fraction of negative eigenvalues (Figure 3c). The differences between the architectures are more prominent early on in the training and there is an indication of a crossover later during training, with less degenerate models early on becoming slightly more degenerate later on as the training performance starts to saturate (Figure 3b). Importantly, the hyper-residual architecture has the highest training speed and the plain architecture has the lowest training speed (Figure 3a), consistent with our hypothesis that the degeneracy of a model increases the training difficulty and skip connections reduce the degeneracy.

## 2.4 TRAINING ACCURACY IS RELATED TO DISTANCE FROM DEGENERATE MANIFOLDS

To establish a more direct relationship between the elimination, overlap and linear dependence singularities discussed earlier on the one hand, and model degeneracy and training difficulty on the other, we exploited the natural variability in training the same model caused by the stochasticity of stochastic gradient descent (SGD) and random initialization. Specifically, we trained 100 plain networks (30 hidden layers, 128 neurons per layer) on CIFAR-100 using different random initializations and random mini-batch selection. Training performance varied widely across runs. We compared the best 10 and the worst 10 runs (measured by mean accuracy over 100 training epochs, Figure 4a). The worst networks were more degenerate (Figure 4b); they were significantly closer to elimination singularities, as measured by the average $l_2$-norm of the incoming weights of their hidden units (Figure 4c); they were significantly closer to overlap singularities (Figure 4d), as measured by the mean correlation between the incoming weights of their hidden units; and their hidden units

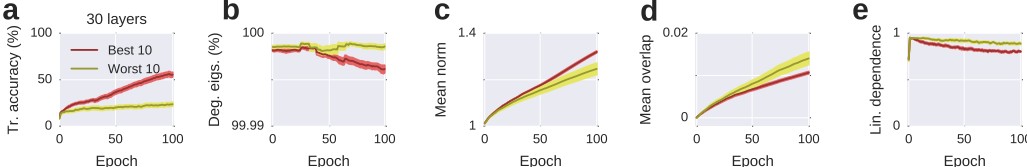

Figure 4: Training accuracy is correlated with distance from degenerate manifolds. (**a**) Training accuracies of the best 10 and the worst 10 plain networks trained on CIFAR-100. (**b**) Estimated fraction of degenerate eigenvalues throughout training. Error bars are standard errors over 10 networks. (**c**) Mean norm of the incoming weight vectors of hidden units. (**d**) Mean overlap of the weight vectors of hidden units. (**e**) Linear dependence between hidden units in the same layer, measured by the fraction of variance explained by the top three eigenmodes of their covariance matrix. These values are averaged over the 30 layers of the network, yielding a single linear dependence score for each network. For a replication of the results shown here for two other datasets, see Figure S7.

were significantly more linearly dependent (Figure 4e), as measured by the mean variance explained by the top three eigenmodes of the covariance matrices of the hidden units in the same layer.

## 2.5 BENEFITS OF SKIP CONNECTIONS AREN'T EXPLAINED BY GOOD INITIALIZATION ALONE

To investigate if the benefits of skip connections can be explained in terms of favorable initialization of the parameters, we introduced a malicious initialization scheme for the residual network by subtracting the identity matrix from the initial weight matrices, $\mathbf{W}_l$. If the benefits of skip connections can be explained primarily by favorable initialization, this malicious initialization would be expected to cancel the effects of skip connections at initialization and hence significantly deteriorate the performance. However, the malicious initialization only had a small adverse effect on the performance of the residual network (Figure 5; ResMalInit), suggesting that the benefits of skip connections cannot be explained by favorable initialization alone. This result reveals a fundamental weakness in previous explanations of the benefits of skip connections based purely on linear models (Hardt & Ma, 2016; Li et al., 2016). In Supplementary Note 5 we show that skip connections do not eliminate the singularities in deep linear networks, but only shift the landscape so that typical initializations are farther from the singularities. Thus, in linear networks, any benefits of skip connections are due entirely to better initialization. In contrast, skip connections genuinely eliminate the singularities in nonlinear networks (Supplementary Note 1). The fact that malicious initialization of the residual network does reduce its performance suggests that "ghosts" of these singularities still exist in the loss landscape of nonlinear networks, but the performance reduction is only slight, suggesting that skip connections alter the landscape around these ghosts to alleviate the learning slow-down that would otherwise take place near them.

## 2.6 ALTERNATIVE WAYS OF ELIMINATING THE SINGULARITIES

If the success of skip connections can be attributed, at least partly, to eliminating singularities, then alternative ways of eliminating them should also improve training. We tested this hypothesis by introducing a particularly simple way of eliminating singularities: for each layer we drew random target biases from a Gaussian distribution, $\mathcal{N}(\mu, \sigma)$, and put an $l_2$-norm penalty on learned biases deviating from those targets. This breaks the permutation symmetry between units and eliminates the overlap singularities. In addition, positive $\mu$ values decrease the average threshold of the units and make the elimination of units less likely (but not impossible), hence reducing the elimination singularities. Decreased thresholds can also increase the dimensionality of the responses in a given layer by reducing the fraction of times different units are identically zero, thereby making them less linearly dependent. Note that setting $\mu = 0$ and $\sigma = 0$ corresponds to the standard $l_2$-norm regularization of the biases, which does not eliminate any of the overlap or elimination singularities. Hence, we expect the performance to be worse in this case than in cases with properly eliminated singularities. On the other hand, although in general, larger values of $\mu$ and $\sigma$ correspond to greater elimination of singularities, the network also has to perform well in the classification task and very large $\mu$, $\sigma$ values might be inconsistent with the latter requirement. Therefore, we expect the performance to be optimal for intermediate values of $\mu$ and $\sigma$. In the experiments reported below, we optimized the hyperparameters $\mu$, $\sigma$, and $\lambda$, i.e. the mean and the standard deviation of the target

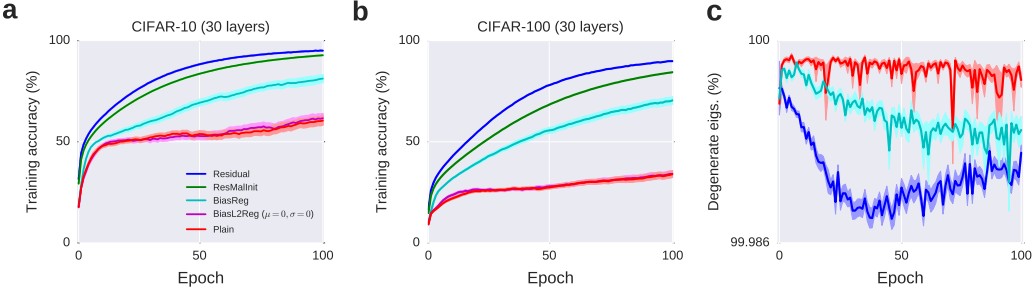

Figure 5: Singularity elimination through bias regularization improves training. (**a-b**) Training accuracy of 30-layer networks on the CIFAR-10 and CIFAR-100 benchmarks. Error bars represent $\pm 1$ SEM over 50 independent runs. (**c**) Estimated fraction of degenerate eigenvalues in the plain, residual and BiasReg networks.

bias distribution and the strength of the bias regularization term, through random search (Bergstra & Bengio, 2012).

We trained 30-layer fully-connected feedforward networks on CIFAR-10 and CIFAR-100 datasets. Figure 5a-b shows the training accuracy of different models on the two datasets. For both datasets, among the models shown in Figure 5, the residual network performs the best and the plain network the worst. Our simple singularity elimination through bias regularization scheme (BiasReg, cyan) significantly improves performance over the plain network. Importantly, the standard $l_2$-norm regularization on the biases (BiasL2Reg ($\mu = 0, \sigma = 0$), magenta) does not improve performance over the plain network. These results are consistent with the singularity elimination hypothesis.

There is still a significant performance gap between our BiasReg network and the residual network despite the fact that both break degeneracies. This can be partly attributed to the fact that the residual network breaks the degeneracies more effectively than the BiasReg network (Figure 5c). Secondly, even in models that completely eliminate the singularities, the learning speed would still depend on the behavior of the gradient norms, and the residual network fares better than the BiasReg network in this respect as well. At the beginning of training, the gradient norms with respect to the layer activities do not diminish in earlier layers of the residual network (Figure 6a, Epoch 0), demonstrating that it effectively solves the vanishing gradients problem (Hochreiter, 1991; Bengio et al., 1994). On the other hand, both in the plain network and in the BiasReg network, the gradient norms decay quickly as one descends from the top of the network. Moreover, as training progresses (Figure 6a, Epochs 1 and 2), the gradient norms are larger for the residual network than for the plain or the BiasReg network. Even for the maliciously initialized residual network, gradients do not decay quickly at the beginning of training and the gradient norms behave similarly to those of the residual network during training (Figure 6a; ResMalInit), suggesting that skip connections boost the gradient norms near the ghosts of the singularities and reduce the learning slow-down that would otherwise take place near them. Adding a single batch normalization layer (Ioffe & Szegedy, 2015) in the middle of the BiasReg network alleviates the vanishing gradients problem for this network and brings its performance closer to that of the residual network (Figure 6a-b; BiasReg+BN).

## 2.7 NON-IDENTITY SKIP CONNECTIONS

If the singularity elimination hypothesis is correct, there should be nothing special about identity skip connections. Skip connections other than identity should lead to training improvements if they eliminate singularities. For the permutation symmetry breaking of the hidden units, ideally the skip connection vector for each unit should disambiguate that unit *maximally* from all other units in that layer. This is because as shown by the analysis in Wei et al. (2008) (Figure 2), even partial overlaps between hidden units significantly slow down learning (near-singularity plateaus). Mathematically, the maximal disambiguation requirement corresponds to an orthogonality condition on the skip connectivity matrix (any full-rank matrix breaks the permutation symmetry, but only orthogonal matrices maximally disambiguate the units). Adding orthogonal vectors to different hidden units is also useful for breaking potential (exact or approximate) linear dependencies between them. We therefore tested random dense orthogonal matrices as skip connectivity matrices. Random dense

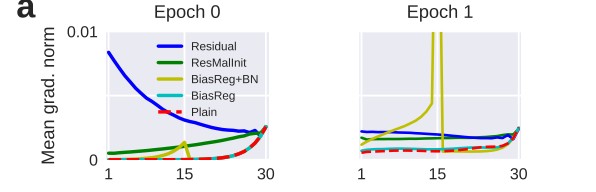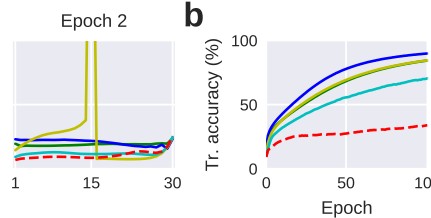

Figure 6: Skip connections effectively deal with the vanishing gradients problem. (**a**) Mean gradient norms with respect to layer activities at the beginning of the first three epochs. Note that the mean gradient norms of Plain and BiasReg networks are almost identical initially. (**b**) Training accuracy of the networks on the CIFAR-100 dataset. The BiasReg network with a single batch normalization layer inserted at layer 15 (BiasReg+BN) is shown in yellow. Its performance approaches the performance of the residual network. The results shown are averages over 50 independent runs. Standard errors are small, hence are not shown for clarity.

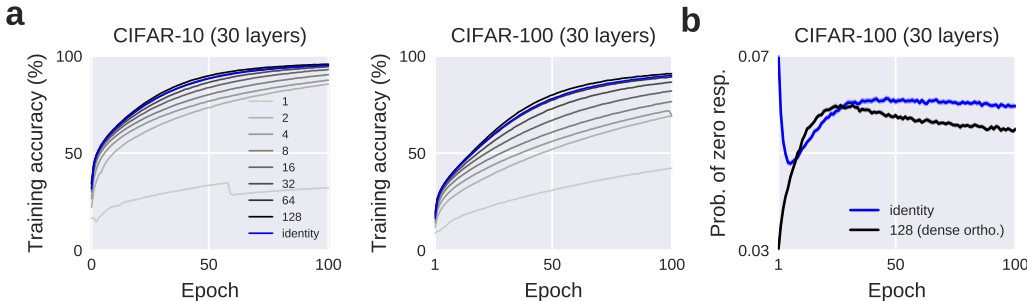

Figure 7: Random dense orthogonal skip connectivity matrices work slightly better than identity skip connections. (**a**) Increasing the non-orthogonality of the skip connectivity matrix reduces the performance (represented by lighter shades of gray). The results shown are averages over 10 independent runs of the simulations. (**b**) Probability of zero responses for residual networks with identity skip connections (blue) and dense random orthogonal skip connections (black), averaged over all hidden units and all training examples.

orthogonal matrices performed slightly better than identity skip connections in both CIFAR-10 and CIFAR-100 datasets (Figure 7a, black vs. blue). This is because, even with skip connections, units can be deactivated for some inputs because of the ReLU nonlinearity (recall that we do not allow skip connections from the input layer). When this happens to a single unit at layer $l$, that unit is effectively eliminated for that subset of inputs, hence eliminating the skip connection to the corresponding unit at layer $l+1$, if the skip connectivity is the identity. This causes a potential elimination singularity for that particular unit. With dense skip connections, however, this possibility is reduced, since all units in the previous layer are used. Moreover, when two distinct units at layer $l$ are deactivated together, the identity skips cannot disambiguate the corresponding units at the next layer, causing a potential overlap singularity. On the other hand, with dense orthogonal skips, because all units at layer $l$ are used, even if some of them are deactivated, the units at layer $l+1$ can still be disambiguated with the remaining active units. Figure 7b confirms for the CIFAR-100 dataset that throughout most of the training, the hidden units of the network with dense orthogonal skip connections have a lower probability of zero responses than those of the network with identity skip connections.

Next, we gradually decreased the degree of "orthogonality" of the skip connectivity matrix to see how the orthogonality of the matrix affects performance. Starting from a random dense orthogonal matrix, we first divided the matrix into two halves and copied the first half to the second half. Starting from $n$ orthonormal vectors, this reduces the number of orthonormal vectors to $n/2$. We continued on like this until the columns of the matrix were repeats of a single unit vector. We predict that as the

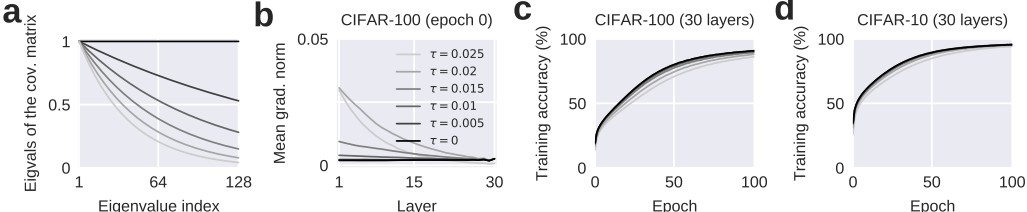

Figure 8: Success of orthogonal skip connections cannot be explained by their ability to deal with vanishing gradients only. (**a**) Eigenvalues of the covariance matrices with different $\tau$ values: $\tau = 0$ corresponds to orthogonal skip connectivity matrices, larger $\tau$ values correspond to less orthogonal matrices. Note that these eigenvalues are the eigenvalues of the covariance matrix of the skip connectivity vectors. The eigenvalue spectra of the skip connectivity matrices are always fixed to be on the unit circle, hence equivalent to that of an orthogonal matrix. (**b**) Mean gradient norms with respect to layer activations at the beginning of training. Gradients do not vanish in less orthogonal skip connectivity matrices. If anything, gradient norms are typically larger with such matrices. (**c-d**) Training accuracies on CIFAR-100 and CIFAR-10.

number of orthonormal vectors in the skip connectivity matrix is decreased, the performance should deteriorate, because both the permutation symmetry-breaking capacity and the linear-dependence-breaking capacity of the skip connectivity matrix are reduced. Figure 7 shows the results for $n = 128$ hidden units. Darker colors correspond to "more orthogonal" matrices (e.g. "128" means all 128 skip vectors are orthonormal to each other, "1" means all 128 vectors are identical). The blue line is the identity skip connectivity. More orthogonal skip connectivity matrices yield better performance, consistent with our hypothesis.

The less orthogonal skip matrices also suffer from the vanishing gradients problem. So, their failure could be partly attributed to the vanishing gradients problem. To control for this effect, we also designed skip connectivity matrices with eigenvalues on the unit circle (hence with eigenvalue spectra equivalent to an orthogonal matrix), but with varying degrees of orthogonality (see Supplementary Note 6 for details). More specifically, the columns (or rows) of an orthogonal matrix are orthonormal to each other, hence the covariance matrix of these vectors is the identity matrix. We designed matrices where this covariance matrix was allowed to have non-zero off-diagonal values, reflecting the fact that the vectors are not orthogonal any more. By controlling the magnitude of the correlations between the vectors, we manipulated the degree of orthogonality of the vectors. We achieved this by setting the eigenvalue spectrum of the covariance matrix to be given by $\lambda_i = \exp(-\tau(i-1))$ where $\lambda_i$ denotes the $i$-th eigenvalue of the covariance matrix and $\tau$ is the parameter that controls the degree of orthogonality: $\tau = 0$ corresponds to the identity covariance matrix, hence to an orthonormal set of vectors, whereas larger values of $\tau$ correspond to gradually more correlated vectors. This orthogonality manipulation was done while fixing the eigenvalue spectrum of the skip connectivity matrix to be on the unit circle. Hence, the effects of this manipulation cannot be attributed to any change in the eigenvalue spectrum, but only to the degree of orthogonality of the skip vectors. The results of this experiment are shown in Figure 8. More orthogonal skip connectivity matrices still perform better than less orthogonal ones (Figure 8c-d), even when their eigenvalue spectrum is fixed and the vanishing gradients problem does not arise (Figure 8b), suggesting that the results of the earlier experiment (Figure 7) cannot be explained solely by the vanishing gradients problem.

## 3    DISCUSSION

In this paper, we proposed a novel explanation for the benefits of skip connections in terms of the elimination of singularities. Our results suggest that elimination of singularities contributes at least partly to the success of skip connections. However, we emphasize that singularity elimination is *not the only factor* explaining the benefits of skip connections. Even in completely non-degenerate models, other independent factors such as the behavior of gradient norms would affect training performance. Indeed, we presented evidence suggesting that skip connections are also quite effective at dealing with the problem of vanishing gradients and not every form of singularity elimination can be expected to be equally good at dealing with such additional problems that beset the training of deep networks.

**Alternative explanations:** Several of our experiments rule out vanishing gradients as *the sole* explanation for training difficulties in deep networks and strongly suggest an independent role for the singularities arising from the non-identifiability of the model. (i) In Figure 4, all nets have the exact same plain architecture and similarly vanishing gradients at the beginning of training, yet they have diverging performances correlated with measures of distance from singular manifolds. (ii) Vanishing gradients cannot explain the difference between identity skips and dense orthogonal skips in Figure 7, because both eliminate vanishing gradients, yet dense orthogonal skips perform better. (iii) In Figure 8, spectrum-equalized non-orthogonal skips often have larger gradient norms, yet worse performance than orthogonal skips. (iv) Vanishing gradients cannot even explain the BiasReg results in Figure 5. The BiasReg and the plain net have almost identical (and vanishing) gradients early on in training (Figure 6a), yet the former has better performance as predicted by the symmetry-breaking hypothesis. (v) Similar results hold for two-layer shallow networks where the problem of vanishing gradients does not arise (Supplementary Note 7). In particular, shallow residual nets are less degenerate and have better accuracy than shallow plain nets; moreover, gradient norms and accuracy are strongly correlated with distance from the overlap manifolds in these shallow nets.

Our malicious initialization experiment with residual nets (Figure 5) suggests that the benefits of skip connections cannot be explained *solely* in terms of well-conditioning or improved initialization either. This result reveals a fundamental weakness in purely linear explanations of the benefits of skip connections (Hardt & Ma, 2016; Li et al., 2016). Unlike in nonlinear nets, improved initialization entirely explains the benefits of skip connections in linear nets (Supplementary Note 5).

A recent paper (Balduzzi et al., 2017) suggested that the loss of spatial structure in the covariance of the gradients, a phenomenon called "shattered gradients", could be partly responsible for training difficulties in deep nonlinear networks. They argued that skip connections alleviate this problem by essentially making the model "more linear". It is easy to see that the shattered gradients problem is distinct from both the vanishing/exploding gradients problem and the degeneracy problems considered in this paper, since shattered gradients arise only in sufficiently non-linear deep networks (linear networks do not shatter gradients), whereas vanishing/exploding gradients, as well as the degeneracies considered here, arise in linear networks too. The relative contribution of each of these distinct problems to training difficulties in deep networks remains to be determined.

**Symmetry-breaking in other architectures:** We only reported results from experiments with fully-connected networks, but we note that limited receptive field sizes and weight sharing between units in a single feature channel in convolutional neural networks also reduce the permutation symmetry in a given layer. The symmetry is not entirely eliminated since although individual units do not have permutation symmetry in this case, feature channels do, but they are far fewer in number than the number of individual units. Similarly, a recent extension of the residual architecture called ResNeXt (Xie et al., 2016) uses parallel, segregated processing streams inside the "bottleneck" blocks, which can again be seen as a way of reducing the permutation symmetry inside the block.

Our method of singularity reduction through bias regularization (BiasReg; Figure 5) can be thought of as indirectly putting a prior over the unit activities. More complicated joint priors over hidden unit responses that favor decorrelated (Cogswell et al., 2015) or clustered (Liao et al., 2016) responses have been proposed before. Although the primary motivation for these regularization schemes was to improve the generalizability or interpretability of the learned representations, they can potentially be understood from a singularity elimination perspective as well. For example, a prior that favors decorrelated responses can facilitate the breaking of permutation symmetries and linear dependencies between hidden units.

Our results lead to an apparent paradox: over-parametrization and redundancy in large neural network models have been argued to make optimization easier. Yet, our results seem to suggest the opposite. However, there is no contradiction here. Any apparent contradiction is due to potential ambiguities in the meanings of the terms "over-parametrization" and "redundancy". The intuition behind the benefits of over-parametrization for optimization is an *increase* in the effective capacity of the model: over-parametrization in this sense leads to a large number of approximately equally good ways of fitting the training data. On the other hand, the degeneracies considered in this paper *reduce* the effective capacity of the model, leading to optimization difficulties. Our results suggest that it could be useful for neural network researchers to pay closer attention to the degeneracies inherent in their models. For better optimization, as a general design principle, we recommend reducing such degeneracies in a model as much as possible. Once the training performance starts to

saturate, however, degeneracies may help the model achieve a better generalization performance. Exploring this trade-off between the harmful and beneficial effects of degeneracies is an interesting direction for future work.

**Acknowledgments:** AEO and XP were supported by the Intelligence Advanced Research Projects Activity (IARPA) via Department of Interior/Interior Business Center (DoI/IBC) contract number D16PC00003. The U.S. Government is authorized to reproduce and distribute reprints for Governmental purposes notwithstanding any copyright annotation thereon. Disclaimer: The views and conclusions contained herein are those of the authors and should not be interpreted as necessarily representing the official policies or endorsements, either expressed or implied, of IARPA, DoI/IBC, or the U.S. Government.

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

## SUPPLEMENTARY MATERIALS

SUPPLEMENTARY NOTE 1: SINGULARITY OF THE HESSIAN IN NON-LINEAR MULTILAYER NETWORKS

Because the cost function can be expressed as a sum over training examples, it is enough to consider the cost for a single example: $E = \frac{1}{2}\|\mathbf{y} - \mathbf{x}_L\|^2 \equiv \frac{1}{2}\mathbf{e}^\top\mathbf{e}$, where $\mathbf{x}_l$ are defined recursively as $\mathbf{x}_l = f(\mathbf{W}_{l-1}\mathbf{x}_{l-1})$ for $l = 1, \ldots, L$. We denote the inputs to units at layer $l$ by the vector $\mathbf{h}_l$: $\mathbf{h}_l = \mathbf{W}_{l-1}\mathbf{x}_{l-1}$. We ignore the biases for simplicity. The derivative of the cost function with respect to a single weight $\mathbf{W}_{l,ij}$ between layers $l$ and $l+1$ is given by:

$$\frac{\partial E}{\partial \mathbf{W}_{l,ij}} = - \begin{bmatrix} 0 \\ \vdots \\ f'(\mathbf{h}_{l+1,i})\mathbf{x}_{l,j} \\ \vdots \\ 0 \end{bmatrix}^\top \mathbf{W}_{l+1}^\top \mathrm{diag}(\mathbf{f}'_{l+2})\mathbf{W}_{l+2}^\top\mathrm{diag}(\mathbf{f}'_{l+3})\cdots\mathbf{W}_{L-1}^\top\mathrm{diag}(\mathbf{f}'_L)\mathbf{e} \quad (1)$$

Now, consider a different connection between the same output unit $i$ at layer $l+1$ and a different input unit $j'$ at layer $l$. The crucial thing to note is that if the units $j$ and $j'$ have the same set of incoming weights, then the derivative of the cost function with respect to $\mathbf{W}_{l,ij}$ becomes identical to its derivative with respect to $\mathbf{W}_{l,ij'}$: $\partial E/\partial \mathbf{W}_{l,ij} = \partial E/\partial \mathbf{W}_{l,ij'}$. This is because in this condition $\mathbf{x}_{l,j'} = \mathbf{x}_{l,j}$ for all possible inputs and all the remaining terms in Equation 1 are independent of the input index $j$. Thus, the columns (or rows) corresponding to the connections $\mathbf{W}_{l,ij}$ and $\mathbf{W}_{l,ij'}$ in the Hessian become identical, making the Hessian degenerate. This is a re-statement of the simple observation that when the units $j$ and $j'$ have the same set of incoming weights, the parameters $\mathbf{W}_{l,ij}$ and $\mathbf{W}_{l,ij'}$ become non-identifiable (only their sum is identifiable). Thus, this corresponds to an *overlap singularity*. A similar argument shows that when a set of units at layer $l$, say units indexed $j, j', j''$ become linearly dependent, the columns of the Hessian corresponding to the weights $\mathbf{W}_{l,ij}$, $\mathbf{W}_{l,ij'}$ and $\mathbf{W}_{l,ij''}$ become linearly dependent as well, thereby making the Hessian singular. Again, this is just a re-statement of the fact that these weights are no longer individually identifiable in this case (only a linear combination of them is identifiable). This corresponds to a *linear dependence singularity*. In non-linear networks, except in certain degenerate cases where the units saturate together, they may never be exactly linearly dependent, but they can be approximately linearly dependent, which makes the Hessian close to singular.

Moreover, it is easy to see from Equation 1 that, when the presynaptic unit $\mathbf{x}_{l,j}$ is always zero, i.e. when that unit is effectively killed, the column (or row) of the Hessian corresponding to the parameter $\mathbf{W}_{l,ij}$ becomes the zero vector for any $i$, and thus the Hessian becomes singular. This is a re-statement of the simple observation that when the unit $\mathbf{x}_{l,j}$ is always zero, its outgoing connections, $\mathbf{W}_{l,ij}$, are no longer identifiable. This corresponds to an *elimination singularity*.

In the residual case, the only thing that changes in Equation 1 is that the factors $\mathbf{W}_k^\top\mathrm{diag}(\mathbf{f}'_{k+1})$ on the right-hand side become $\mathbf{W}_k^\top\mathrm{diag}(\mathbf{f}'_{k+1}) + \mathbf{I}$ where $\mathbf{I}$ is an identity matrix of the appropriate size.

The overlap singularities are eliminated, because $\mathbf{x}_{l,j'}$ and $\mathbf{x}_{l,j}$ cannot be the same for all possible inputs in the residual case (even when the adjustable incoming weights of these units are identical). Similarly, elimination singularities are also eliminated, because $\mathbf{x}_{l,j}$ cannot be identically zero for all possible inputs (even when the adjustable incoming weights of this unit are all zero), assuming that the corresponding unit at the previous layer $\mathbf{x}_{l-1,j}$ is not always zero, which, in turn, is guaranteed with an identity skip connection if $\mathbf{x}_{l-2,j}$ is not always zero etc., all the way down to the first hidden layer. Any linear dependence between $\mathbf{x}_{l,j}$, $\mathbf{x}_{l,j'}$ and $\mathbf{x}_{l,j''}$ is also eliminated by adding linearly independent inputs to them, assuming again that the corresponding units in the previous layer are linearly independent.

SUPPLEMENTARY NOTE 2: SIMULATION DETAILS

In Figure 3, for the skip connections between non-adjacent layers in the hyper-residual networks, i.e. $\mathbf{Q}_k$, we used matrices of the type labeled "32" in Figure 7, i.e. matrices consisting of four copies of a set of 32 orthonormal vectors. We found that these matrices performed slightly better than orthogonal matrices.

We augmented the training data in both CIFAR-10 and CIFAR-100 by adding reflected versions of each training image, i.e. their mirror images. This yields a total of 100000 training images for both datasets. The test data were not augmented, consisting of 10000 images in both cases. We used the standard splits of the data into training and test sets.

For the BiasReg network of Figures 5-6, random hyperparameter search returned the following values for the target bias distributions: $\mu = 0.51$, $\sigma = 0.96$ for CIFAR-10 and $\mu = 0.91$, $\sigma = 0.03$ for CIFAR-100.

The toy model shown in Figure 2b-c consists of the simulation of Equations 3.7 and 3.9 in Wei et al. (2008). The toy model shown in Figure 2e is the simulation of learning dynamics in a network with 3 input, 3 hidden and 3 output units, parametrized in terms of the norms and unit-vector directions of $J_a - J_b - J_c$, $J_a + J_b - J_c$, $J_c$, and the output weights. A teacher model with random parameters is first chosen and a large set of "training data" is generated from the teacher model. Then the gradient flow fields with respect to the two parameters $m = ||J_a + J_b - J_c||$ and $||J_c||$ are plotted with the assumption that the remaining parameters are already at their optima (a similar assumption was made in the analysis of Wei et al. (2008)). We empirically confirmed that the flow field is generic.

SUPPLEMENTARY NOTE 3: ESTIMATING THE EIGENVALUE SPECTRAL DENSITY OF THE HESSIAN IN DEEP NETWORKS

We use Skilling's moment matching method (Skilling, 1989) to estimate the eigenvalue spectra of the Hessian. We first estimate the first few non-central moments of the density by computing $m_k = \frac{1}{N}\mathbf{r}^\top \mathbf{H}^k \mathbf{r}$ where $\mathbf{r}$ is a random vector drawn from the standard multivariate Gaussian with zero mean and identity covariance, $\mathbf{H}$ is the Hessian and $N$ is the dimensionality of the parameter space. Because the standard multivariate Gaussian is rotationally symmetric and the Hessian is a symmetric matrix, it is easy to show that $m_k$ gives an unbiased estimate of the $k$-th moment of the spectral density:

$$m_k = \frac{1}{N}\mathbf{r}^\top \mathbf{H}^k \mathbf{r} = \frac{1}{N}\sum_{i=1}^{N} \tilde{\mathbf{r}}_i^2 \lambda_i^k \to \int p(\lambda)\lambda^k d\lambda \quad \text{as} \quad N \to \infty \tag{2}$$

where $\lambda_i$ are the eigenvalues of the Hessian, and $p(\lambda)$ is the spectral density of the Hessian as $N \to \infty$. In Equation 2, we make use of the fact that $\tilde{\mathbf{r}}_i^2$ are random variables with expected value 1.

Despite appearances, the products in $m_k$ do not require the computation of the Hessian explicitly and can instead be computed efficiently as follows:

$$\mathbf{v}_0 = \mathbf{r}, \qquad \mathbf{v}_k = \mathbf{H}\mathbf{v}_{k-1} \quad k = 1, \ldots, K \tag{3}$$

where the Hessian times vector computation can be performed without computing the Hessian explicitly through Pearlmutter's $\mathcal{R}$-operator (Pearlmutter, 1994). In terms of the vectors $\mathbf{v}_k$, the estimates of the moments are given by the following:

$$m_{2k} = \frac{1}{N}\mathbf{v}_k^\top \mathbf{v}_k, \qquad m_{2k+1} = \frac{1}{N}\mathbf{v}_k^\top \mathbf{v}_{k+1} \tag{4}$$

For the results shown in Figure 3, we use 20-layer fully-connected feedforward networks and the number of parameters is $N = 709652$. For the remaining simulations, we use 30-layer fully-connected networks and the number of parameters is $N = 874772$.

We estimate the first four moments of the Hessian and fit the estimated moments with a parametric density model. The parametric density model we use is a mixture of a narrow Gaussian distribution (to capture the bulk of the density) and a skew-normal distribution (to capture the tails):

$$q(\lambda) = w\mathcal{SN}(\lambda; \xi, \omega, \alpha) + (1 - w)\mathcal{N}(\lambda; 0, \sigma = 0.001) \tag{5}$$

with 4 parameters in total: the mixture weight $w$, and the location $\xi$, scale $\omega$ and shape $\alpha$ parameters of the skew-normal distribution. We fix the parameters of the Gaussian component to $\mu = 0$ and $\sigma = 0.001$. Since the densities are heavy-tailed, the moments are dominated by the tail behavior of the model, hence the fits are not very sensitive to the precise choice of the parameters of the Gaussian component. The moments of our model can be computed in closed-form. We had difficulty fitting the parameters of the model with gradient-based methods, hence we used a simple grid search method instead. The ranges searched over for each parameter was as follows. $w$: logarithmically spaced between $10^{-9}$ and $10^{-3}$; $\alpha$: linearly spaced between $-50$ and $50$; $\xi$: linearly spaced between $-10$ and $10$; $\omega$: logarithmically spaced between $10^{-1}$ and $10^3$. 100 parameters were evaluated along each parameter dimension for a total of $10^8$ parameter configurations evaluated.

The estimated moments ranged over several orders of magnitude. To make sure that the optimization gave roughly equal weight to fitting each moment, we minimized a normalized objective function:

$$\mathcal{L}(w, \alpha, \xi, \omega) = \sum_{k=1}^{4} \frac{|\hat{m}_k(w, \alpha, \xi, \omega) - m_k|}{|m_k|} \tag{6}$$

where $\hat{m}_k(w, \alpha, \xi, \omega)$ is the model-derived estimate of the $k$-th moment.

SUPPLEMENTARY NOTE 4: VALIDATION OF THE RESULTS WITH SMALLER NETWORKS

Here, we validate our main results for smaller, numerically tractable networks. The networks in this section are 10-layer fully-connected feedforward networks. The networks are trained on CIFAR-100. The input dimensionality is reduced from 3072 to 128 through PCA. In what follows, we calculate the fraction of degenerate eigenvalues by counting the number of eigenvalues inside a small window of size $0.2$ around $0$, and the fraction of negative eigenvalues by the number of eigenvalues to the left of this window.

We first compare residual networks with plain networks (Figure S1). The networks here have 16 hidden units in each layer yielding a total of $4852$ parameters. This is small enough that we can calculate all eigenvalues of the Hessian numerically. We observe that residual networks have better training and test performance (Figure S1a-b); they are less degenerate (Figure S1d) and have more negative eigenvalues than plain networks (Figure S1c). These results are consistent with the results reported in Figure 3 for deeper and larger networks.

Next, we validate the results reported in Figure 4 by running 400 independent plain networks and comparing the best-performing 40 with the worst-performing 40 among them (Figure S2). Again, the networks here have 16 hidden units in each layer with a total of $4852$ parameters. We observe that the best networks are less degenerate (Figure S2d) and have more negative eigenvalues than the worst networks (Figure S2c). Moreover, the hidden units of the best networks have less overlap (Figure S2f), and, at least initially during training, have slightly larger weight norms than the worst-performing networks (Figure S2e). Again, these results are all consistent with those reported in Figure 4 for deeper and larger networks.

Finally, using numerically tractable plain networks, we also tested whether we could reliably estimate the fractions of degenerate and negative eigenvalues with our mixture model. Just as we do for the larger networks, we first fit the mixture model to the first four moments of the spectral density estimated with the method of Skilling (1989). We then estimate the fraction of degenerate and negative eigenvalues from the fitted mixture model and compare these estimates with those obtained from the numerically calculated actual eigenvalues. Because for the larger networks, the networks were found to be highly degenerate, we restrict the analysis here to conditions where the fraction of degenerate eigenvalues was at least $99.8\%$. We used 10-layer plain networks with 32 hidden

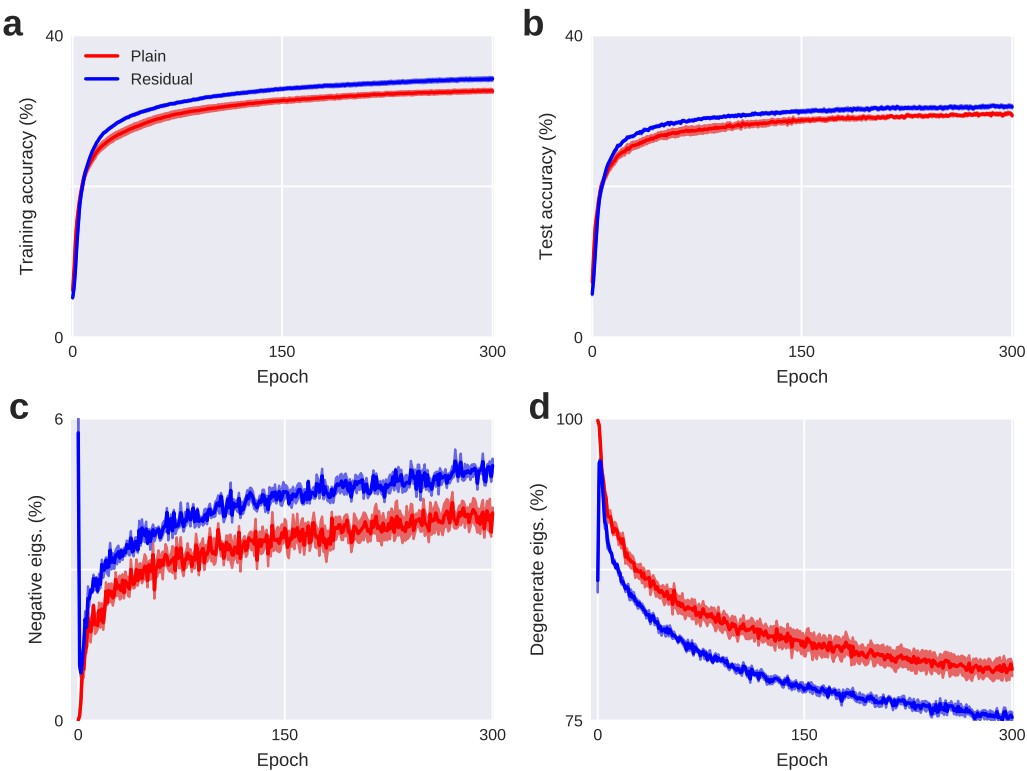

Figure S1: Validation of the results with 10-layer plain and residual networks trained on CIFAR-100. (**a-b**) Training and test accuracy. (**c-d**) Fraction of negative and degenerate eigenvalues throughout training. The results are averages over 4 independent runs ±1 standard errors.

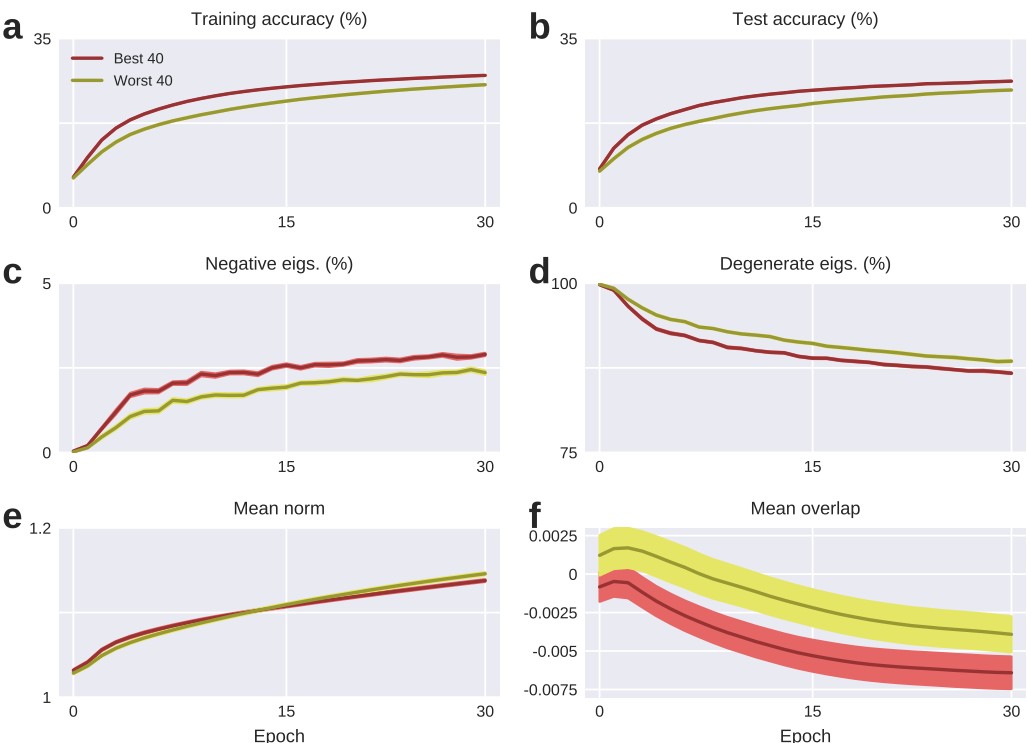

Figure S2: Validation of the results with 400 10-layer plain networks with 16 hidden units in each layer (4852 parameters total) trained on CIFAR-100. We compare the best 40 networks with the worst 40 networks, as in Figure 4. (**a-b**) Training and test accuracy. (**c-d**) Fraction of negative and degenerate eigenvalues throughout training. Better performing networks are less degenerate and have more negative eigenvalues. (**e**) Mean norms of the incoming weight vectors of the hidden units. (**f**) Mean overlaps of the hidden units as measured by the mean correlation between their incoming weight vectors. The results are averages over 40 best or worst runs $\pm 1$ standard errors.

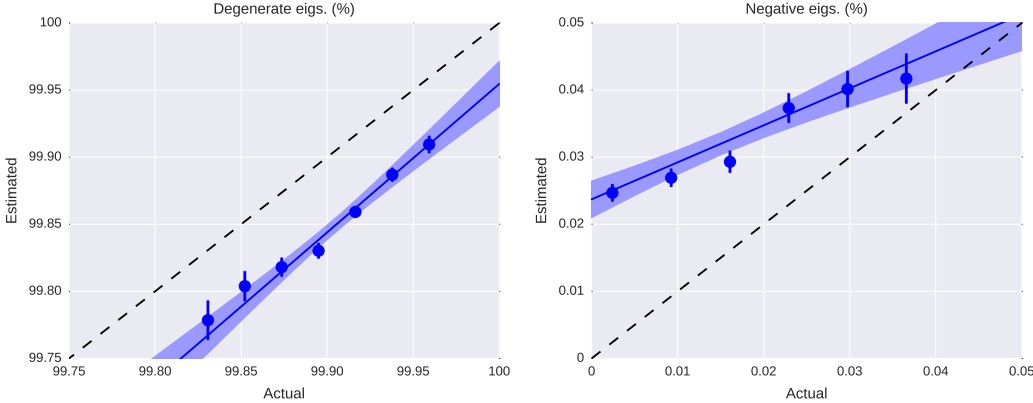

Figure S3: For 10-layer plain networks with 32 hidden units in each layer (14292 parameters total), estimates obtained from the mixture model slightly underestimate the fraction of degenerate eigenvalues, and overestimate the fraction of negative eigenvalues; however, there is a highly significant linear relationship between the actual values and the estimates. (**a**) Actual vs. estimated fraction of degenerate eigenvalues. (**b**) Actual vs. estimated fraction of negative eigenvalues for the same networks. Dashed line shows the identity line. Dots and errorbars represent means and standard errors of estimates in different bins; the solid lines and the shaded regions represent the linear regression fits and the 95% confidence intervals.

units in each layer (with a total of 14292 parameters) for this analysis. We observe that, at least for these small networks, the mixture model usually underestimates the fraction of degenerate eigenvalues and overestimates the fraction of negative eigenvalues. However, there is a highly significant positive correlation between the actual and estimated fractions (Figure S3).

SUPPLEMENTARY NOTE 5: DYNAMICS OF LEARNING IN LINEAR NETWORKS WITH SKIP CONNECTIONS

To get a better analytic understanding of the effects of skip connections on the learning dynamics, we turn to linear networks. In an $L$-layer linear plain network, the input-output mapping is given by (again ignoring the biases for simplicity):

$$\mathbf{x}_L = \mathbf{W}_{L-1}\mathbf{W}_{L-2}\dots\mathbf{W}_1\mathbf{x}_1 \tag{7}$$

where $\mathbf{x}_1$ and $\mathbf{x}_L$ are the input and output vectors, respectively. In linear residual networks with identity skip connections between adjacent layers, the input-output mapping becomes:

$$\mathbf{x}_L = (\mathbf{W}_{L-1} + \mathbf{I})(\mathbf{W}_{L-2} + \mathbf{I})\dots(\mathbf{W}_1 + \mathbf{I})\mathbf{x}_1 \tag{8}$$

Finally, in hyper-residual linear networks where all skip connection matrices are assumed to be the identity, the input-output mapping is given by:

$$\mathbf{x}_L = \Big(\mathbf{W}_{L-1} + (L-1)\mathbf{I}\Big)\Big(\mathbf{W}_{L-2} + (L-2)\mathbf{I}\Big)\dots\Big(\mathbf{W}_1 + \mathbf{I}\Big)\mathbf{x}_1 \tag{9}$$

In the derivations to follow, we do not have to assume that the connectivity matrices are square matrices. If they are rectangular matrices, the identity matrix $\mathbf{I}$ should be interpreted as a rectangular identity matrix of the appropriate size. This corresponds to zero-padding the layers when they are not the same size, as is usually done in practice.

**Three-layer networks:** Dynamics of learning in plain linear networks with no skip connections was analyzed in Saxe et al. (2013). For a three-layer network ($L = 3$), the learning dynamics can be expressed by the following differential equations (Saxe et al., 2013):

$$\tau\frac{d}{dt}a^\alpha = (s_\alpha - a^\alpha \cdot b^\alpha)b^\alpha - \sum_{\gamma \neq \alpha}(a^\alpha \cdot b^\gamma)b^\gamma \tag{10}$$

$$\tau\frac{d}{dt}b^\alpha = (s_\alpha - a^\alpha \cdot b^\alpha)a^\alpha - \sum_{\gamma \neq \alpha}(a^\gamma \cdot b^\alpha)a^\gamma \tag{11}$$

Here $a^\alpha$ and $b^\alpha$ are $n$-dimensional column vectors (where $n$ is the number of hidden units) connecting the hidden layer to the $\alpha$-th input and output modes, respectively, of the input-output correlation matrix and $s_\alpha$ is the corresponding singular value (see Saxe et al. (2013) for further details). The first term on the right-hand side of Equations 10-11 facilitates cooperation between $a^\alpha$ and $b^\alpha$ corresponding to the same input-output mode $\alpha$, while the second term encourages competition between vectors corresponding to different modes.

In the simplest scenario where there are only two input and output modes, the learning dynamics of Equations 10, 11 reduces to:

$$\frac{d}{dt}a^1 = (s_1 - a^1 \cdot b^1)b^1 - (a^1 \cdot b^2)b^2 \tag{12}$$

$$\frac{d}{dt}a^2 = (s_2 - a^2 \cdot b^2)b^2 - (a^2 \cdot b^1)b^1 \tag{13}$$

$$\frac{d}{dt}b^1 = (s_1 - a^1 \cdot b^1)a^1 - (a^1 \cdot b^2)a^2 \tag{14}$$

$$\frac{d}{dt}b^2 = (s_2 - a^2 \cdot b^2)a^2 - (a^2 \cdot b^1)a^1 \tag{15}$$

How does adding skip connections between adjacent layers change the learning dynamics? Considering again a three-layer network ($L = 3$) with only two input and output modes, a straightforward extension of Equations 12-15 shows that the learning dynamics changes as follows:

$$\frac{d}{dt}a^1 = \Big[s_1 - (a^1 + v^1) \cdot (b^1 + u^1)\Big](b^1 + u^1) - \Big[(a^1 + v^1) \cdot (b^2 + u^2)\Big](b^2 + u^2) \tag{16}$$

$$\frac{d}{dt}a^2 = \Big[s_2 - (a^2 + v^2) \cdot (b^2 + u^2)\Big](b^2 + u^2) - \Big[(a^2 + v^2) \cdot (b^1 + u^1)\Big](b^1 + u^1) \tag{17}$$

$$\frac{d}{dt}b^1 = \Big[s_1 - (a^1 + v^1) \cdot (b^1 + u^1)\Big](a^1 + v^1) - \Big[(a^1 + v^1) \cdot (b^2 + u^2)\Big](a^2 + v^2) \tag{18}$$

$$\frac{d}{dt}b^2 = \Big[s_2 - (a^2 + v^2) \cdot (b^2 + u^2)\Big](a^2 + v^2) - \Big[(a^2 + v^2) \cdot (b^1 + u^1)\Big](a^1 + v^1) \tag{19}$$

where $u^1$ and $u^2$ are orthonormal vectors (similarly for $v^1$ and $v^2$). The derivation proceeds essentially identically to the corresponding derivation for plain networks in Saxe et al. (2013). The only differences are: (i) we substitute the plain weight matrices $\mathbf{W}_l$ with their residual counterparts $\mathbf{W}_l + \mathbf{I}$ and (ii) when changing the basis from the canonical basis for the weight matrices $\mathbf{W}_1, \mathbf{W}_2$ to the input and output modes of the input-output correlation matrix, $\mathbf{U}$ and $\mathbf{V}$, we note that:

$$\mathbf{W}_2 + \mathbf{I} = \mathbf{U}\overline{\mathbf{W}}_2 + \mathbf{U}\mathbf{U}^\top = \mathbf{U}(\overline{\mathbf{W}}_2 + \mathbf{U}^\top) \tag{20}$$

$$\mathbf{W}_1 + \mathbf{I} = \overline{\mathbf{W}}_1\mathbf{V}^\top + \mathbf{V}\mathbf{V}^\top = (\overline{\mathbf{W}}_1 + \mathbf{V})\mathbf{V}^\top \tag{21}$$

where $\mathbf{U}$ and $\mathbf{V}$ are orthogonal matrices and the vectors $a^\alpha$, $b^\alpha$, $u^\alpha$ and $v^\alpha$ in Equations 16-19 correspond to the $\alpha$-th columns of the matrices $\overline{\mathbf{W}}_1$, $\overline{\mathbf{W}}_2^\top$, $\mathbf{U}$ and $\mathbf{V}$, respectively.

Figure S4 shows, for two different initializations, the evolution of the variables $a^1$ and $a^2$ in plain and residual networks with two input-output modes and two hidden units. When the variables are initialized to small random values, the dynamics in the plain network initially evolves slowly (Figure S4a, blue); whereas it is much faster in the residual network (Figure S4a, red). This effect is attributable to two factors. First, the added orthonormal vectors $u^\alpha$ and $v^\alpha$ increase the initial velocity of the variables in the residual network. Second, even when we equalize the initial norms of the vectors, $a^\alpha$ and $a^\alpha + v^\alpha$ (and those of the vectors $b^\alpha$ and $b^\alpha + u^\alpha$) in the plain and the residual networks, respectively, we still observe an advantage for the residual network (Figure S4b), because the cooperative and competitive terms are orthogonal to each other in the residual network (or close to orthogonal, depending on the initialization of $a^\alpha$ and $b^\alpha$; see right-hand side of Equations 16-19), whereas in the plain network they are not necessarily orthogonal and hence can cancel each other (Equations 12-15), thus slowing down convergence.

**Singularity of the Hessian in linear three-layer networks:** The dynamics in Equations 10, 11 can be interpreted as gradient descent on the following energy function:

$$E = \frac{1}{2\tau} \sum_\alpha (s_\alpha - a^\alpha \cdot b^\alpha)^2 + \frac{1}{2\tau} \sum_{\alpha \neq \beta} (a^\alpha \cdot b^\beta)^2 \tag{22}$$

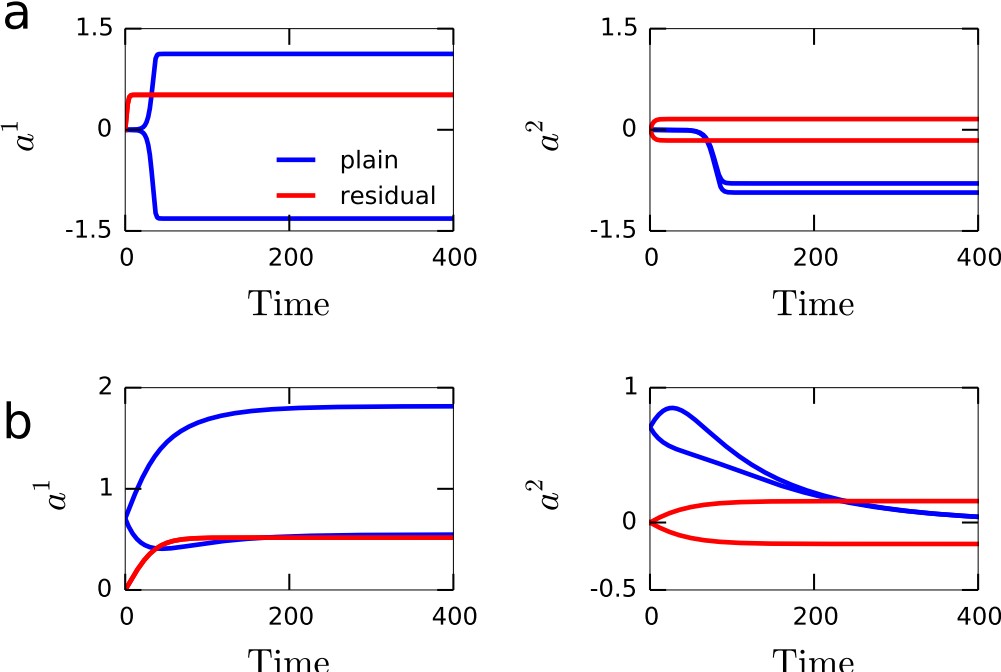

Figure S4: Evolution of $a^1$ and $a^2$ in linear plain and residual networks (evolution of $b^1$ and $b^2$ proceeds similarly). The weights converge faster in residual networks. Simulation details are as follows: the number of hidden units is 2 (the two solid lines for each color represent the weights associated with the two hidden nodes, e.g. $a_1^1$ and $a_2^1$ on the left), the singular values are $s_1 = 3.0$, $s_2 = 1.5$. For the residual network, $u_1 = v_1 = [1/\sqrt{2}, 1/\sqrt{2}]^\top$ and $u_2 = v_2 = [1/\sqrt{2}, -1/\sqrt{2}]^\top$. In (**a**), the weights of both plain and residual networks are initialized to random values drawn from a Gaussian with zero mean and standard deviation of $0.0001$. The learning rate was set to $0.1$. In (**b**), the weights of the plain network are initialized as follows: the vectors $a^1$ and $a^2$ are initialized to $[1/\sqrt{2}, 1/\sqrt{2}]^\top$ and the vectors $b^1$ and $b^2$ are initialized to $[1/\sqrt{2}, -1/\sqrt{2}]^\top$; the weights of the residual network are all initialized to zero, thus equalizing the initial norms of the vectors $a^\alpha$ and $a^\alpha + v^\alpha$ (and those of the vectors $b^\alpha$ and $b^\alpha + u^\alpha$) between the plain and residual networks. The residual network still converges faster than the plain network. In (**b**), the learning rate was set to $0.01$ to make the different convergence rates of the two networks more visible.

This energy function is invariant to a (simultaneous) permutation of the elements of the vectors $a^\alpha$ and $b^\alpha$ for all $\alpha$. This causes degenerate manifolds in the landscape. Specifically, for the permutation symmetry of hidden units, these manifolds are the hyperplanes $a_i^\alpha = a_j^\alpha \ \forall \alpha$, for each pair of hidden units $i$, $j$ (similarly, the hyperplanes $b_i^\alpha = b_j^\alpha \ \forall \alpha$) that make the model non-identifiable. Formally, these correspond to the singularities of the Hessian or the Fisher information matrix. Indeed, we shall quickly check below that when $a_i^\alpha = a_j^\alpha \ \forall \alpha$ for any pair of hidden units $i$, $j$, the Hessian becomes singular (*overlap singularities*). The Hessian also has additional singularities at the hyperplanes $a_i^\alpha = 0 \ \forall \alpha$ for any $i$ and at $b_i^\alpha = 0 \ \forall \alpha$ for any $i$ (*elimination singularities*).

Starting from the energy function in Equation 22 and taking the derivative with respect to a single input-to-hidden layer weight, $a_i^\alpha$:

$$\frac{\partial E}{\partial a_i^\alpha} = -(s_\alpha - a^\alpha \cdot b^\alpha)b_i^\alpha + \sum_{\beta \neq \alpha}(a^\alpha \cdot b^\beta)b_i^\beta \tag{23}$$

and the second derivatives are as follows:

$$\frac{\partial^2 E}{\partial (a_i^\alpha)^2} = (b_i^\alpha)^2 + \sum_{\beta \neq \alpha}(b_i^\beta)^2 = \sum_\beta (b_i^\beta)^2 \tag{24}$$

$$\frac{\partial^2 E}{\partial a_i^\alpha \partial a_j^\alpha} = b_j^\alpha b_i^\alpha + \sum_{\beta \neq \alpha} b_j^\beta b_i^\beta = \sum_\beta b_i^\beta b_j^\beta \tag{25}$$

Note that the second derivatives are independent of mode index $\alpha$, reflecting the fact that the energy function is invariant to a permutation of the mode indices. Furthermore, when $b_i^\beta = b_j^\beta$ for all $\beta$, the columns in the Hessian corresponding to $a_i^\alpha$ and $a_j^\alpha$ become identical, causing an additional degeneracy reflecting the non-identifiability of $a_i^\alpha$ and $a_j^\alpha$. A similar derivation establishes that $a_i^\beta = a_j^\beta$ for all $\beta$ also leads to a degeneracy in the Hessian, this time reflecting the non-identifiability of $b_i^\alpha$ and $b_j^\alpha$. These correspond to the overlap singularities.

In addition, it is easy to see from Equations 24, 25 that when $b_i^\alpha = 0 \ \forall \alpha$, the right-hand sides of both equations become identically zero, reflecting the non-identifiability of $a_i^\alpha$ for all $\alpha$. A similar derivation shows that when $a_i^\alpha = 0 \ \forall \alpha$, the columns of the Hessian corresponding to $b_i^\alpha$ become identically zero for all $\alpha$, this time reflecting the non-identifiability of $b_i^\alpha$ for all $\alpha$. These correspond to the elimination singularities.

When we add skip connections between adjacent layers, i.e. in the residual architecture, the energy function changes as follows:

$$E = \frac{1}{2}\sum_\alpha (s_\alpha - (a^\alpha + v^\alpha) \cdot (b^\alpha + u^\alpha))^2 + \frac{1}{2}\sum_{\alpha \neq \beta}((a^\alpha + v^\alpha) \cdot (b^\beta + u^\beta))^2 \tag{26}$$

and straightforward algebra yields the following second derivatives:

$$\frac{\partial^2 E}{\partial (a_i^\alpha)^2} = \sum_\beta (b_i^\beta + u_i^\beta)^2 \tag{27}$$

$$\frac{\partial^2 E}{\partial a_i^\alpha \partial a_j^\alpha} = \sum_\beta (b_i^\beta + u_i^\beta)(b_j^\beta + u_j^\beta) \tag{28}$$

Unlike in the plain network, setting $b_i^\beta = b_j^\beta$ for all $\beta$, or setting $b_i^\alpha = 0 \ \forall \alpha$, does not lead to a degeneracy here, thanks to the orthogonal skip vectors $u^\beta$. However, this just shifts the locations of the singularities. In particular, the residual network suffers from the same overlap and elimination singularities as the plain network when we make the following change of variables: $b^\beta \to b^\beta - u^\beta$ and $a^\beta \to a^\beta - v^\beta$.

**Networks with more than three-layers:** As shown in Saxe et al. (2013), in linear networks with more than a single hidden layer, assuming that there are orthogonal matrices $\mathbf{R}_l$ and $\mathbf{R}_{l+1}$ for each layer $l$ that diagonalize the initial weight matrix of the corresponding layer (i.e. $\mathbf{R}_{l+1}^\top \mathbf{W}_l(0)\mathbf{R}_l = \mathbf{D}_l$ is a diagonal matrix), dynamics of different singular modes decouple from each other and each

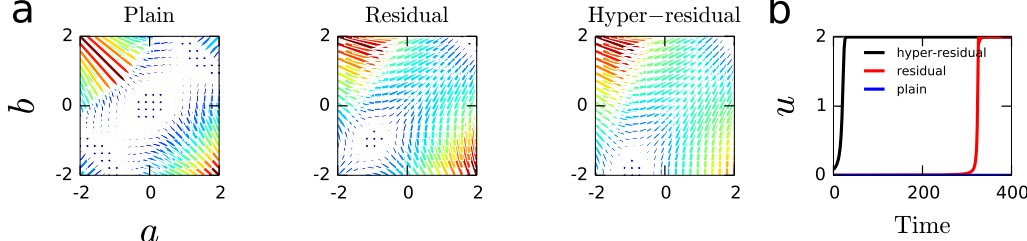

Figure S5: **(a)** Phase portraits for three-layer plain, residual and hyper-residual linear networks. **(b)** Evolution of $u = \prod_{l=1}^{N_l-1} a_l$ for 10-layer plain, residual and hyper-residual linear networks. In the plain network, $u$ did not converge to its asymptotic value $s$ within the simulated time window.

mode $\alpha$ evolves according to gradient descent dynamics in an energy landscape described by (Saxe et al., 2013):

$$E_{plain} = \frac{1}{2\tau}\left(s_\alpha - \prod_{l=1}^{N_l-1} a_l^\alpha\right)^2 \tag{29}$$

where $a_l^\alpha$ can be interpreted as the strength of mode $\alpha$ at layer $l$ and $N_l$ is the total number of layers. In residual networks, assuming further that the orthogonal matrices $\mathbf{R}_l$ satisfy $\mathbf{R}_{l+1}^\top \mathbf{R}_l = \mathbf{I}$, the energy function changes to:

$$E_{res} = \frac{1}{2\tau}\left(s_\alpha - \prod_{l=1}^{N_l-1} (a_l^\alpha + 1)\right)^2 \tag{30}$$

and in hyper-residual networks, it is:

$$E_{hyperres} = \frac{1}{2\tau}\left(s_\alpha - \prod_{l=1}^{N_l-1} (a_l^\alpha + l)\right)^2 \tag{31}$$

Figure S5a illustrates the effect of skip connections on the phase portrait of a three layer network. The two axes, $a$ and $b$, represent the mode strength variables for $l = 1$ and $l = 2$, respectively: i.e. $a \equiv a_1^\alpha$ and $b \equiv a_2^\alpha$. The plain network has a saddle point at $(0, 0)$ (Figure S5a; left). The dynamics around this point is slow, hence starting from small random values causes initially very slow learning. The network funnels the dynamics through the unstable manifold $a = b$ to the stable hyperbolic solution corresponding to $ab = s$. Identity skip connections between adjacent layers in the residual architecture move the saddle point to $(-1, -1)$ (Figure S5a; middle). This speeds up the dynamics around the origin, but not as much as in the hyper-residual architecture where the saddle point is moved further away from the origin and the main diagonal to $(-1, -2)$ (Figure S5a; right). We found these effects to be more pronounced in deeper networks. Figure S5b shows the dynamics of learning in 10-layer linear networks, demonstrating a clear advantage for the residual architecture over the plain architecture and for the hyper-residual architecture over the residual architecture.

**Singularity of the Hessian in reduced linear multilayer networks with skip connections:** The derivative of the cost function of a linear multilayer residual network (Equation 30) with respect to the mode strength variable at layer $i$, $a_i$, is given by (suppressing the mode index $\alpha$ and taking $\tau = 1$):

$$\frac{\partial E}{\partial a_i} = -(s - u)\prod_{l\neq i}(a_l + 1) \tag{32}$$

and the second derivatives are:

$$\frac{\partial^2 E}{\partial a_i^2} = \left[\prod_{l\neq i}(a_l + 1)\right]^2 \tag{33}$$

$$\frac{\partial^2 E}{\partial a_i \partial a_k} = \left[2\prod_l(a_l + 1) - s\right]\prod_{l\neq i,k}(a_l + 1) \tag{34}$$

It is easy to check that the columns (or rows) corresponding to $a_i$ and $a_j$ in the Hessian become identical when $a_i = a_j$, making the Hessian degenerate. The hyper-residual architecture does not eliminate these degeneracies but shifts them to different locations in the parameter space by adding distinct constants to $a_i$ and $a_j$ (and to all other variables).

SUPPLEMENTARY NOTE 6: DESIGNING SKIP CONNECTIVITY MATRICES WITH VARYING DEGREES OF ORTHOGONALITY AND WITH EIGENVALUES ON THE UNIT CIRCLE

We generated the covariance matrix of the eigenvectors by $\mathbf{S} = \mathbf{Q}\mathbf{\Lambda}\mathbf{Q}^\top$, where $\mathbf{Q}$ is a random orthogonal matrix and $\mathbf{\Lambda}$ is the diagonal matrix of eigenvalues, $\mathbf{\Lambda}_{ii} = \exp(-\tau(i-1))$, as explained in the main text. We find the correlation matrix through $\mathbf{R} = \mathbf{D}^{-1/2}\mathbf{S}\mathbf{D}^{-1/2}$ where $\mathbf{D}$ is the diagonal matrix of the variances: i.e. $\mathbf{D}_{ii} = \mathbf{S}_{ii}$. We take the Cholesky decomposition of the correlation matrix, $\mathbf{R} = \mathbf{T}\mathbf{T}^\top$. Then the designed skip connectivity matrix is given by $\mathbf{\Sigma} = \mathbf{T}\mathbf{U}\mathbf{L}\mathbf{U}^{-1}\mathbf{T}^{-1}$, where $\mathbf{L}$ and $\mathbf{U}$ are the matrices of eigenvalues and eigenvectors of another randomly generated orthogonal matrix, $\mathbf{O}$: i.e. $\mathbf{O} = \mathbf{U}\mathbf{L}\mathbf{U}^\top$. With this construction, $\mathbf{\Sigma}$ has the same eigenvalue spectrum as $\mathbf{O}$, however the eigenvectors of $\mathbf{\Sigma}$ are linear combinations of the eigenvectors of $\mathbf{O}$ such that their correlation matrix is given by $\mathbf{R}$. Thus, the eigenvectors of $\mathbf{\Sigma}$ are not orthogonal to each other unless $\tau = 0$. Larger values of $\tau$ yield more correlated, hence less orthogonal, eigenvectors.

SUPPLEMENTARY NOTE 7: VALIDATION OF THE RESULTS WITH SHALLOW NETWORKS

To further demonstrate the generality of our results and the independence of the problem of singularities from the vanishing gradients problem in optimization, we performed an experiment with shallow plain and residual networks with only two hidden layers and 16 units in each hidden layer. Because we do not allow skip connections from the input layer, a network with two hidden layers is the shallowest network we can use to compare the plain and residual architectures. Figure S6 shows the results of this experiment. The residual network performs slightly better both on the training and test data (Figure S6a-b); it is less degenerate (Figure S6d) and has more negative eigenvalues (Figure S6c); it has larger gradients (Figure S6e) —note that the gradients in the plain network do not vanish even at the beginning of training— and its hidden units have less overlap than the plain network (Figure S6f). Moreover, the gradient norms closely track the mean overlap between the hidden units and the degeneracy of the network (Figure S6d-f) throughout training. These results suggest that the degeneracies caused by the overlaps of hidden units slow down learning, consistent with our symmetry-breaking hypothesis and with the results from larger networks.

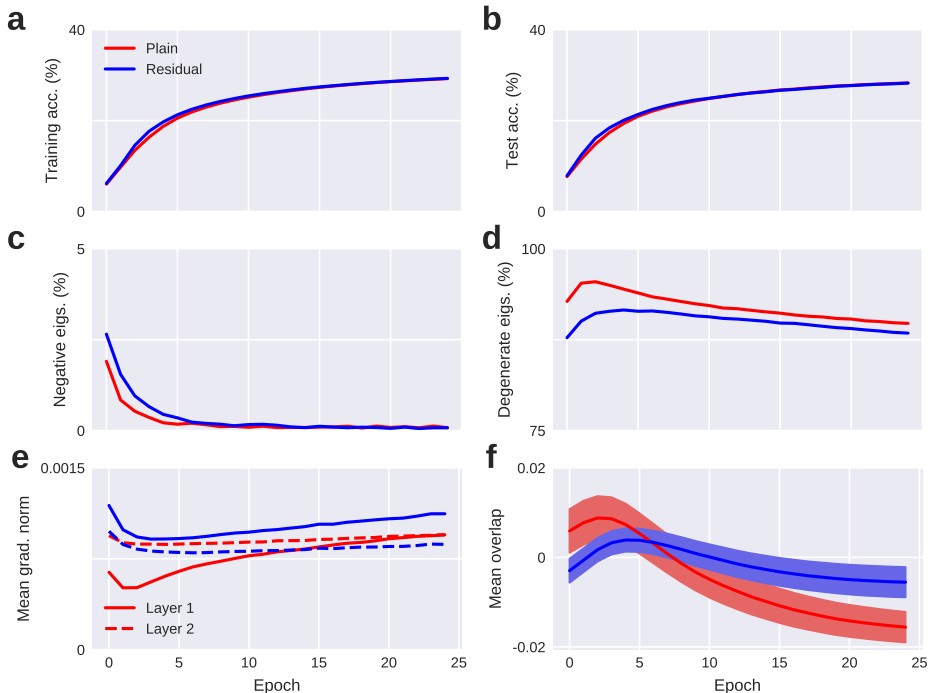

Figure S6: Main results hold for two-layer shallow nets trained on CIFAR-100. (**a-b**) Training and test accuracies. Residual nets perform slightly better. (**c-d**) Fraction of negative and degenerate eigenvalues. Residual nets are less degenerate. (**e**) Mean gradient norms with respect to the two layer activations throughout training. (**f**) Mean overlap for the second hidden layer units, measured as the mean correlation between the incoming weights of the hidden units. Results in (a-e) are averages over 16 independent runs; error bars are small, hence not shown for clarity. In (f), error bars represent standard errors.

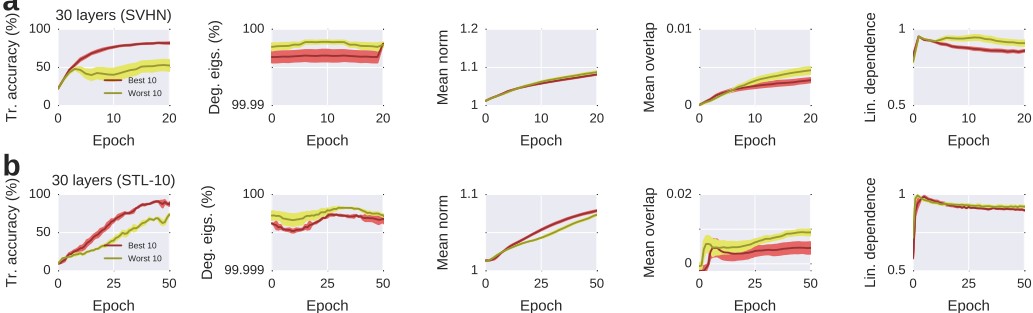

Figure S7: Replication of the results reported in Figure 4 for (**a**) the Street View House Numbers (SVHN) dataset and (**b**) the STL-10 dataset.

