# OpenReview forum: "Skip Connections Eliminate Singularities"
_ICLR.cc/2018/Conference — Accept (Poster)_

### Official Review · AnonReviewer1 · 2017-11-22
**Thorough study, useful result**

**Rating:** 8
**Confidence:** 3

**Review:**

The authors show that two types of singularities impede learning in deep neural networks: elimination singularities (where a unit is effectively shut off by a loss of input or output weights, or by an overly-strong negative bias), and overlap singularities, where two or more units have very similar input or output weights. They then demonstrate that skip connections can reduce the prevalence of these singularities, and thus speed up learning.

The analysis is thorough: the authors explore alternative methods of reducing the singularities, and explore the skip connection properties that more strongly reduce the singularities, and make observations consistent with their overarching claims.

I have no major criticisms.

One suggestion for future work would be to provide a procedure for users to tailor their skip connection matrices to maximize learning speed and efficacy. The authors could then use this procedure to make highly trainable networks, and show that on test (not training) data, the resultant network leads to high performance.

---

### Official Review · AnonReviewer2 · 2017-11-27
**Well-written paper examining how skip connections influence training dynamics in deep nets.**

**Rating:** 8
**Confidence:** 3

**Review:**

Paper examines the use of skip connections (including residual layers) in deep networks as a way of alleviating two perceived difficulties in training: 1) when a neuron does not contain any information, and 2) when two neurons in a layer compute the same function. Both of these cases lead to singularities in the Hessian matrix, and this work includes a number of experiments showing the effect of skip connections on the Hessian during training.

This is a significant and timely topic. While I may not be the best one to judge the originality of this work, I appreciated how the authors presented clear and concise arguments with experiments to back up their claims.

---

### Official Review · AnonReviewer3 · 2017-11-29

**Rating:** 6
**Confidence:** 4

**Review:**

This paper proposes to explain the benefits of skip connections in terms of eliminating the singularities of the loss function. The discussion is largely based on a sequence of experiments, some of which are interesting and insightful. The discussion here can be useful for other researchers.

My main concern is that the result here is purely empirical, with no concrete theoretical justification. What the experiments reveal is an empirical correlation between the Eigval index and training accuracy, which can be caused by lots of reasons (and cofounders), and does not necessarily establish a causal relation. Therefore, i found many of the discussion to be questionable. I would love to see more solid theoretical discussion to justify the hypothesis proposed in this paper.

Do you have a sense how accurate is the estimation of the tail probabilities of the eigenvalues? Because the whole paper is based on the approximation of the eigval indexes, it is critical to exam the estimation is accurate enough to draw the conclusions in the paper.

All the conclusions are based on one or two datasets. Could you consider testing the result on more different datasets to verify if the results are generalizable?

---

> ### Author Response · Authors · 2018-01-03
> **Response to reviewer 3**
>
> We agree with the reviewer that a theoretically more rigorous analysis of the results presented in our paper would be desirable. However, both because the paper is already quite long and because the analysis requested by the reviewer is not straightforward even in simplified models, we would like to leave this for future work. We would also like to point out that the idea that degeneracies cause training difficulties in neural networks is not wholly without theoretical precedent. Although the models they deal with are highly simplified models, prior work by Amari and colleagues, which we discuss in our paper, already established this connection. Similarly, as also discussed in our paper, Saxe et al. (2013) showed that randomly initialized linear networks become increasingly degenerate with depth and they identified this as the source of the training difficulties in such deep networks.
>
> Supplementary Note 4 validates our methods for quantifying degeneracies in smaller, numerically tractable networks. The results from these smaller networks agree with the results from the larger networks presented in the main text. For these smaller networks, the mixture model slightly underestimated the fraction of degenerate eigenvalues and overestimated the fraction of degenerate eigenvalues. However, in both cases, there was a highly significant linear relationship between the estimated and actual fractions. This suggests that inferences drawn from the mixture model about the relative degeneracy of models should be reliable, while inferences about the exact values of the eigenvalue degeneracy or negativity of the models should be made more cautiously.
>
> As requested, we tested the results shown in Figure 4 in two more image recognition datasets (SVHN and STL-10). The results from these new datasets are presented in the supplementary Figure S7. These results are in agreement with the ones from CIFAR-100 shown in Figure 4.

---

### Public Comment · (anonymous) · 2017-12-03
**Skip connections formed by orthogonal and idempotent transformations**

This paper presents a good analysis on analyzing skip connections for training deep NNs. I noticed that the paper, "Orthogonal and Idempotent Transformations for Learning Deep Neural Networks" by Jingdong Wang, Yajie Xing, Kexin Zhang, Cha Zhang, provided two methods for designing skip connections, which might be related to "orthogonality" mentioned in this paper. Is there any possible discussion?

---

> ### Author Response · Authors · 2018-01-03
> **Orthogonal skip connections**
>
> Yes, the orthogonal skip connectivity proposed in the preprint mentioned is the same as the one we propose in our submission. We note that this preprint appeared several months after our preprint.

---

### Author Response · Authors · 2018-01-03
**Updates**

 We thank the reviewers for their comments and positive feedback.

Since the original submission, we have noticed that another degeneracy that could potentially be significant in training deep networks is linear dependence between hidden units. The paper is updated with a discussion and some new results regarding this degeneracy. Results in Figure 4 are replicated for two more datasets (SVHN and STL-10, supplementary Figure S7). Several more minor changes are also made in the revision to improve the presentation.

---

### Decision · Program_Chairs · 2018-01-29
**ICLR 2018 Conference Acceptance Decision**

**Decision:**

Accept (Poster)

**Comment:**

pros:
* novel explanation: skip connections <--> singualrities
* thorough analysis
* significant topic in understanding deep nets

cons:
* more rigorous theoretical analysis would be better

overall, the committee feels this paper would be interesting to have at ICLR.